# Does TRODAT-1 SPECT Uptake Correlate with Cerebrospinal Fluid α-Synuclein Levels in Mid-Stage Parkinson’s Disease?

**DOI:** 10.3390/biomedicines11020296

**Published:** 2023-01-20

**Authors:** Artur M. Coutinho, Maria Gabriela Ghilardi, Ana Carolina P. Campos, Elba Etchebehere, Fernanda C. Fonoff, Rubens G. Cury, Rosana L. Pagano, Raquel C. R. Martinez, Erich T. Fonoff

**Affiliations:** 1Division of Neuroscience, Hospital Sírio-Libanês, Sao Paulo 01308-060, SP, Brazil; 2Laboratory of Nuclear Medicine (LIM 43), Department of Radiology and Oncology, Faculdade de Medicina (FMUSP), Universidade de Sao Paulo, Sao Paulo 05403-010, SP, Brazil; 3Division of Nuclear Medicine and PET/CT, Hospital Sírio-Libanês, Sao Paulo 01308-050, SP, Brazil; 4Department of Neurology, Faculdade de Medicina (FMUSP), Universidade de Sao Paulo, Sao Paulo 05403-010, SP, Brazil; 5Division of Nuclear Medicine, University of Campinas (UNICAMP), Campinas 13083-888, SP, Brazil; 6LIM/23—Institute of Psychiatry, Faculdade de Medicina (FMUSP), Universidade de Sao Paulo, Sao Paulo 05403-903, SP, Brazil

**Keywords:** Parkinson’s disease, mid-stage Parkinson, TRODAT-1, α-synuclein, laterality, dopamine

## Abstract

Background: Parkinson’s disease (PD) is characterized by a progressive loss of nigrostriatal dopaminergic neurons with impaired motor and non-motor symptoms. It has been suggested that motor asymmetry could be caused due to an imbalance in dopamine levels, as visualized by dopamine transporter single emission computed tomography test (DAT-SPECT), which might be related to indirect measures of neurodegeneration, evaluated by the Montreal Cognitive Assessment (MOCA) and α-synuclein levels in the cerebrospinal fluid (CSF). Therefore, this study aimed to understand the correlation between disease laterality, DAT-SPECT, cognition, and α-synuclein levels in PD. Methods: A total of 28 patients in the moderate-advanced stage of PD were subjected to neurological evaluation, TRODAT-1-SPECT/CT imaging, MOCA, and quantification of the levels of α-synuclein. Results: We found that α-synuclein in the CSF was correlated with global cognition (positive correlation, r^2^ = 0.3, *p* = 0.05) and DAT-SPECT concentration in the putamen (positive correlation, r^2^ = 0.4, *p* = 0.005), and striatum (positive correlation, r^2^ = 0.2, *p* = 0.03), thus working as a neurodegenerative biomarker. No other correlations were found between DAT-SPECT, CSF α-synuclein, and cognition, thus suggesting that they may be lost with disease progression. Conclusions: Our data highlight the importance of understanding the dysfunction of the dopaminergic system in the basal ganglia and its complex interactions in modulating cognition.

## 1. Introduction

The worldwide prevalence of Parkinson’s disease (PD) has increased significantly over the last decade due to an increase in longevity [1]. Initially, PD was well recognized by the progressive loss of dopaminergic neurons within the nigrostriatal pathway [2], which compromises the motor circuitry and causes classic motor and non-motor symptoms [3,4]. The motor symptoms of PD are often initially present on one side of the body, and a certain asymmetry may persist even after bilateral motor dysfunction [5,6]. One of the many explanations for the motor asymmetry observed in these patients is the imbalance of dopamine levels between brain hemispheres [7,8]. However, dopamine imbalance has been shown to influence many non-motor symptoms of PD [9], including cognitive deficits [10,11].

The increase in α-synuclein in Lewy bodies is an essential pathological hallmark of PD [12,13,14]. Therefore, its expression in cerebrospinal fluid (CSF) has been introduced as a possible biomarker for PD [15,16]. However, to date, its use to determine diagnosis or prognosis is still under investigation [17]. Regarding non-motor symptoms, cognitive impairment is usually slow and insidious and should be included in the evaluation of PD biomarkers [18].

Regarding imaging investigations, the use of dopamine transporter (DAT) single emission computed tomography (SPECT) with 123I-ioflupane has shown 98% sensitivity and specificity for detecting nigrostriatal cell loss in patients with PD [19]. It is used to investigate the neurodegenerative nature of PD. DAT-SPECT using the less explored 99mTc-TRODAT tracer could also work as a proxy for gross dopamine loss, imbalance between hemispheres, and an indirect measure of neurodegeneration, which could be related to non-motor symptoms.

Therefore, a possible correlation between disease laterality, DAT-SPECT uptake, cognition, and CSF α-synuclein in PD should be thoroughly studied. Hence, in this study, we aimed to investigate these relationships in patients with PD to better understand the neurobiology of PD.

## 2. Materials and Methods

### 2.1. Patients

This is a prospective clinical trial. The study was conducted in accordance with the Declaration of Helsinki and approved by the Ethics Committee of the Hospital Sirio-Libanês (CAAE: 02857412.9.0000.5461) and was inserted into the Clinical Trials database with the following ClinicalTrial.gov identifier: NCT02647372. Twenty-eight patients diagnosed with PD were evaluated by the Department of Functional Neurosurgery at the Hospital das Clínicas, assisted by the public health system. The inclusion criterion was mid-stage PD according to the established international guidelines [20]. The exclusion criteria were impairment of balance, swallowing, and speech disorders related to PD, anatomical abnormalities, and pre-existing uncontrolled medical conditions, as shown in Table 1. All the patients included in this study signed a written informed consent form.

### 2.2. Outcomes

#### 2.2.1. 99mTc-TRODAT-1-SPECT Imaging

SPECT imaging was performed using a hybrid multislice (6 channel) SPECT/CT camera (Symbia™ T Series SPECT/CT, Siemens Healthineers, Munich, German) in awake patients four hours after infection with 814–1036 MBq of 99mTc-TRODAT-1. Volumes of interest (VOIs) were drawn with the aid of the CT images in both the striatum and their nuclei (the caudate nucleus and the putamen) as well as in the occipital cortex (OCC) and quantified as follows: VOI of the target area—VOI in the OCC/VOI in the OCC. The numeric results from the left and right striatum and from each nucleus were used in the analyses.

#### 2.2.2. CSF α-Synuclein Expression

The CSF of all the patients was obtained by lumbar puncture from vertebral bodies L3–L5, collected in polypropylene tubes that were divided into aliquots, and stored at −80 °C. The procedure was performed during the afternoon. The samples were then subjected to α-synuclein concentration analyses using the Luminex method (#HNS1MAG-95K) according to the manufacturer’s recommendations. Briefly, samples were incubated overnight with specific magnetic beads, followed by detection antibodies and a streptavidin-phycoerythrin solution. All incubation steps were performed in an orbital shaker (room temperature at 250 rpm). Finally, the assay was analyzed using the MAGPIX^®^ software (MAGPIX System, Luminex Corporation, Austin, TX, USA).

#### 2.2.3. Montreal Cognitive Assessment (MoCA) Score

All patients included in the study were evaluated by the same experienced neuropsychologist using the MoCA scale [21] in order to standardize the evaluation and assess the levels of cognition in PD. Briefly, the MoCA scale properly translated to the country of the patients evaluated: short-term memory recall task, visuospatial abilities, executive functions, phonemic fluency, attention, concentration and work memory, language, abstract reasoning, and orientation to time and place. A MoCa score of ≥26 points is considered normal cognition, and cognitive impairment is a score <26, as determined for the general population.

#### 2.2.4. Neurological Assessment

All patients were evaluated by a movement disorders specialist in PD. The disease laterality was assessed based on the patient’s self-report of the side that first showed motor symptoms.

## 3. Results

### 3.1. General Description of Individuals with PD

From a total of 30 patients, two were excluded due to anatomical abnormalities and pre-existing, uncontrolled medical conditions. We included 28 individuals (see Figure 1; 36% female; mean age, 58 years) with a mean disease duration of 10 years. Of note, these individuals were considered to be in the moderate-advanced stage of PD (Table 2).

### 3.2. Disease Laterality, Dopamine Transporter Uptake, and CSF α-Synuclein

In patients who first showed symptoms on the left side of the body, we observed no significant difference regarding the DAT uptake of the right and left sides in the caudate, putamen, or total striatum (F(1.27) = 2.004, *p* = 0.1739; Figure 2a). Interestingly, in these patients, uptake in the left putamen (r = 0.4; *p* = 0.005; Figure 2b) and left total striatum (r = 0.2; *p* = 0.023; Figure 2c) was negatively correlated with the expression of α-synuclein in the CSF. However, in individuals that first showed symptoms on the right side of the body, we found no significant asymmetry in the right and left caudate, putamen, or total striatum (F(1.27) = 3.01, *p* = 0.234; Figure 2d) nor a significant correlation with CSF α-synuclein levels (r = 0.1; *p* = 0.3; Figure 2e and r = 0.1; *p* = 0.3; Figure 2f).

Individuals that initially exhibited symptoms on the right side of the body tended to have a lower MoCA score than those that presented on the left (*p* = 0.047; Figure 3a). There was a weak positive correlation between the MoCA score and CSF α-synuclein (r = 0.3; *p* = 0.05; Figure 3b). No major correlations were found when pooling together patients regardless of the side of the body with initial motor impairment.

## 4. Discussion

The present study contradicts previous published literature that showed a dopaminergic asymmetrical profile in the initially affected body [7,8]. A classical relation is shown with the onset of the first motor symptoms of PD occurring when 80% of striatal dopamine cells are lost [3] and DAT-SPECT uptake reflecting disease severity [8]. Therefore, the lack of asymmetry may be explained by the longer disease duration in our cohort, which may also explain the lack of a correlation between DAT density and cognition. Our data suggest that the hemispheric imbalance of striatal dopamine is lost with disease progression.

In left-sided individuals with PD, a correlation was observed between DAT density on the ipsilateral side of the body with initial symptoms and CSF α-synuclein expression. Initial reduced uptake on DAT-SPECT is usually contralateral to the most affected side, which turns bilateral with disease progression [8]. Thus, changes in the ipsilateral side of the symptoms may be the last to be detected, granting a more comprehensive range of uptake among individuals and consequently being able to show correlations. However, this can only be confirmed in a larger cohort, including individuals with early- to late-stage PD and possibly unimpaired individuals.

The differences between left- and right-side PD onset have been contradictory. While no difference has been shown regarding neuropsychological profiles or motor deficits in early PD [20], left-side PD onset individuals demonstrated poorer visual memory, while right-side PD onset individuals showed lower verbal memory [22]. Right-sided patients exhibited reduced cognitive function [23]. To better understand the role of the side in disease presentation, it is important to evaluate subgroups of PD and their association with the main characteristics of PD, such as rigid-akinetic or tremor [24].

Most patients included in our study had MoCA scores in the normal range, which were positively correlated with CSF α-synuclein levels. This suggests that the evolution of cognitive impairment is linked to α-synuclein pathology in PD, confirming that patients with mid-stage PD usually have minor impairments in cognition [25,26]. Additionally, it could suggest that CSF α-synuclein might be a more sensitive measure of neurodegeneration than cognitive tests. CSF α-synuclein levels may reflect PD pathology and rise before other tests become abnormal [27].

Finally, we showed that CSF α-synuclein is linked to global cognition in mid-stage PD and to nigrostriatal dopamine levels measured with DAT-SPECT in the less impaired basal ganglia, thus working as a neurodegenerative biomarker. The correlations between DAT-SPECT, CSF α-synuclein, and cognition may be lost with disease progression.

## 5. Conclusions

Our data highlight the importance of understanding the dysfunction of the dopaminergic system in the basal ganglia and their complex interactions to modulate cognition.

## Figures and Tables

**Figure 1 biomedicines-11-00296-f001:**
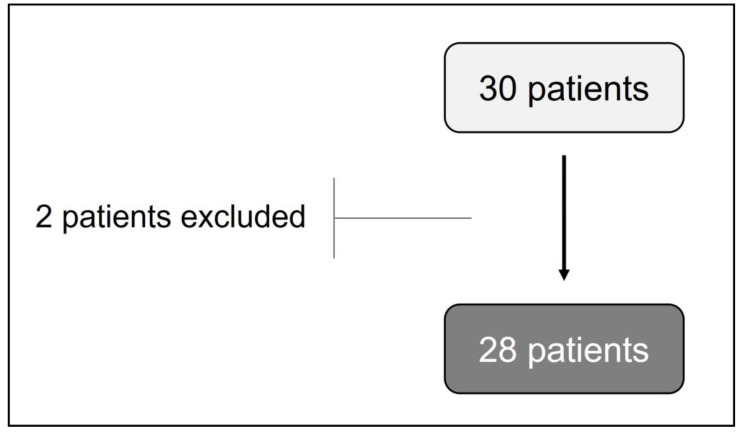
Flowchart of patients included in the analysis. From a total of 30 patients, two were excluded due to anatomical abnormalities and pre-existing uncontrolled medical conditions.

**Figure 2 biomedicines-11-00296-f002:**
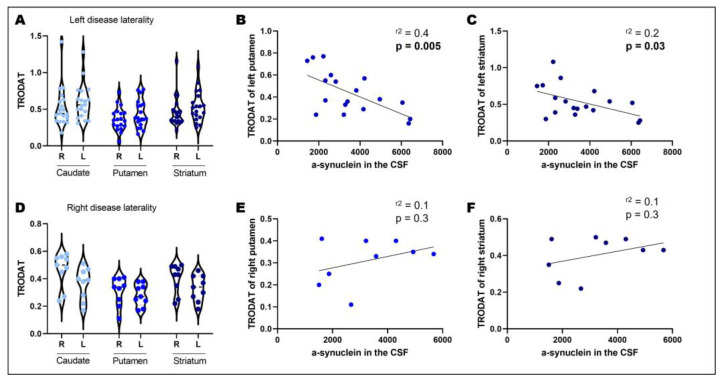
Disease laterality, dopamine transporter uptake, and CSF α-synuclein. 99mTc-TRODAT-1 quantification performed in the caudate, putamen, and whole striatum of PD individuals with left disease laterality showed no difference between hemispheres (**A**). Spearman’s rank correlation coefficient between 99mTc-TRODAT-1 quantification in the left putamen and α-synuclein in the CSF shows a positive correlation, indicating that patients with increased expression of α-synuclein in the CSF also have increased 99mTc-TRODAT-1 levels in the left putamen (**B**). Spearman’s rank correlation coefficient between 99mTc-TRODAT-1 quantification in the left striatum and α-synuclein in the CSF shows a positive correlation, indicating that patients with increased expression of α-synuclein in the CSF also have increased 99mTc-TRODAT-1 levels in the left striatum (**C**). 99mTc-TRODAT-1 quantification performed in the caudate, putamen, and whole striatum of PD individuals with right disease laterality showed no difference between hemispheres (**D**). Spearman’s rank correlation coefficient between 99mTc-TRODAT-1 quantification in the right putamen and α-synuclein in the CSF shows a positive correlation, indicating that patients with increased expression of α-synuclein in the CSF also have increased 99mTc-TRODAT-1 levels in the right putamen (**E**). Spearman’s rank correlation coefficient between 99mTc-TRODAT-1 quantification in the right striatum and α-synuclein in the CSF shows a positive correlation, indicating that patients with increased expression of α-synuclein in the CSF also have increased 99mTc-TRODAT-1 levels in the left striatum (**F**). CSF: cerebrospinal fluid; R: right hemisphere; L: left hemisphere.

**Figure 3 biomedicines-11-00296-f003:**
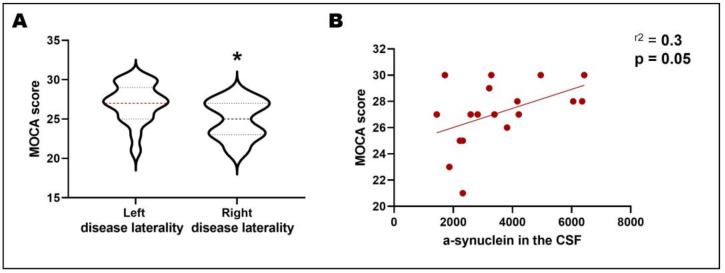
Montreal Cognitive Assessment (MoCA), disease laterality, and CSF α-synuclein. MoCA scores compare left and right disease laterality in individuals. A score of ≥26 points is considered normal cognition, and cognitive impairment is a score of <26 (**A**). The Spearman’s rank correlation coefficient between the MoCA score and the α-synuclein expression in the CSF (**B**). CSF: cerebrospinal fluid. *: *p* < 0.05.

**Table 1 biomedicines-11-00296-t001:** Inclusion and exclusion criteria.

Inclusion Criteria	Exclusion Criteria
Mid-stage PD	Impairment balance
Hoehn and Yahr score: 2.0–3.0	Swallowing
After 12 h of off medication	Speech disorders
	Anatomical anormalities
	Pre-existing, uncontrolled medical conditions


**Table 2 biomedicines-11-00296-t002:** Cohort description.

**Gender**		
Female (*n*, %)	10	35.7%
Male (*n*, %)	18	64.3%
**Age**		
Mean (SD)	57.86	8.84
Median (Min–Max)	58.57	41–76
**Disease duration**		
Mean (SD)	10.18	4.47
Median (Min–Max)	10	1–20
**Disease severity**		
Mean (SD)	42.86	8.52
Median (Min–Max)	40.5	25–57

## Data Availability

The datasets generated and/or analyzed during the current study are available from the corresponding author upon reasonable request in the REDCap database (https://redcap.iephsl.org.br) (last accessed on 9 December 2022).

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
