# Peer review of "Does TRODAT-1 SPECT Uptake Correlate with Cerebrospinal Fluid α-Synuclein Levels in Mid-Stage Parkinson’s Disease?"

_biomedicines, 2023, doi:10.3390/biomedicines11020296_

Round 1

Reviewer 1 Report

The authors investigated the correlation between DAT-SPECT, cognition, and α-synuclein levels in PD. However, there are a few major comments that need to be addressed below:

Page 1; Line 20: Delete "n" and may insert "A"

Page 1; Line 22-23: It would be great if the statistically significant positive correlation values of α-synuclein and DAT-SPECT in the CSF concentration should be used in the abstract.

Page 2; Line 21-22: It would be great if the inclusion and exclusion criteria should be given in a tabular form.

Page 2; Line 37-39: CSF α-synuclein expression analysis should be brief even using previous methods or the manufacturer's kit.

Page 2; Line 40: Montreal Cognitive Assessment (MoCA) score evaluation should be brief even using previous methods.

The results showed little explanation for the given figures, it should be explained more on the basis of figure legends.

Author Response

São Paulo, January 05th, 2023

Manuscript biomedicines-2121994

To reviewers of Biomedicines,

We would like to thank the reviewers for the careful analysis of the manuscript. We appreciate the comments, which have considerably improved our manuscript. We have addressed all the concerns raised by the reviewers with much attention. Please find below the responses to the reviewers’ comments within this letter and in the revised manuscript. The changes were added with “track changes” tool as per requested.

Sincerely,

Raquel C R Martinez, PhD

biomedicines-2121994

Title: Does TRODAT-1 SPECT uptake correlate with cerebrospinal fluid α -synuclein levels in mid-stage Parkinson’s disease?

Author’s name: Coutinho et al.

The manuscript was reviewed according to the comments and the changes are listed below:

REVIEWER 1

The authors investigated the correlation between DAT-SPECT, cognition, and α-synuclein levels in PD. However, there are a few major comments that need to be addressed below:

  • Page 1; Line 20: Delete "n" and may insert "A".

Answer: Thank you for bringing that to our attention, we modified as suggested.

  • Page 1; Line 22-23: It would be great if the statistically significant positive correlation values of α-synuclein and DAT-SPECT in the CSF concentration should be used in the abstract.

Answer: We added as per requested.

  • Page 2; Line 21-22: It would be great if the inclusion and exclusion criteria should be given in a tabular form.

Answer: We greatly appreciate the suggestion and added the information in as Table 1.

  • Page 2; Line 37-39: CSF α-synuclein expression analysis should be brief even using previous methods or the manufacturer's kit.

Answer: We agree with the reviewer and added a brief protocol procedure as per requested.

  • Page 2; Line 40: Montreal Cognitive Assessment (MoCA) score evaluation should be brief even using previous methods.

Answer: We agree with the reviewer and added a brief explanation regarding MoCA per requested.

  • The results showed little explanation for the given figures, it should be explained more on the basis of figure legends.

Answer: We are very sorry for this, and we added further explanation of the results in the legends figures.

Reviewer 2 Report

Review of a manuscript “Does TRODAT-1 SPECT uptake correlate with cerebrospinal fluid α -synuclein levels in mid-stage Parkinson’s disease?” by Artur Martins N Coutinho and coauthors submitted to “Biomedicines”

Parkinson’s disease is a prevalent and severe neurodegenerative disease for which currently there is no treatment modifying the course of the disorder. Furthermore, there is no reliable biomarker for early identification of the beginning of the disease, and the number of patients with the disorder is growing. Thus, this important to investigate molecular and cellular mechanisms of the disease. The authors used dopamine transporter single emission computed tomography test (DAT-SPECT) to study the imbalance in dopamine levels, and to find possible relationship with other measures of neurodegeneration, and with α-synuclein levels in the cerebrospinal fluid. This is an important field of biomedical science and the results of the manuscript will be interesting to the readers of “Biomedicines”. The following corrections and additions should be made:

Abstract

The following sentence should be corrected “(4) Conclusions: Our data highlight the importance of understanding the dysfunction of the dopaminergic system in the basal ganglia and its complex interactions to modulate cognition” should be corrected as follows:” (4) Conclusion: Our data highlight the importance of understanding the dysfunction of the dopaminergic system in the basal ganglia and its complex interactions in modulation of cognition”

Introduction

The following sentence:” The worldwide prevalence of Parkinson’s disease (PD) has increased significantly over the last decade owing to an increase in longevity [1]” should be corrected as follows:” The worldwide prevalence of Parkinson’s disease (PD) has increased significantly over the last decade due to an increase in human longevity [1]”.

After the sentence “:However, to date, its use to determine diagnosis or prognosis is still under investigation.” The authors should add a citation to a recent article:”Biomarkers in Parkinson’s Disease”. Chapter in a book Peplow P.V., Martinez B., Gennarelli T.A. (eds) Neurodegenerative Diseases Biomarkers. 2022. Neuromethods, vol 173. pp 155-180. Humana, New York, NY. https://link.springer.com/protocol/10.1007/978-1-0716-1712-0_7”

“Therefore, a possible correlation between disease laterality, DAT-SPECT uptake, cognition, and CSF α-synuclein in PD is lacking” should be corrected as follows:” Therefore, a possible correlation between disease laterality, DAT-SPECT uptake, cognition, and CSF α-synuclein in PD should be thoroughly studied”

“…and was inserted at the Clinical Trials: NCT02647372” should be rewritten as follows :” and was inserted at the Clinical Trials with the following ClinicalTrials.gov Identifier :NCT02647372”

Results

3.1 General description of individuals with PD

“We included 28 individuals, see Figure 1 (26% female)…” The authors should explain the contradiction between 26% female shown in the text and 35.7% in the Table 1.

 Discussion

“The present study did not confirm a dopamine imbalance between the brain hemispheres in mid-to late-stage PD [7,8].” The sense of this statement is not clear. Do the authors mean that dopamine imbalance between the brain hemispheres in mid-to late-stage PD was previously detected in these two studies? [7,8]. And their findings in this study contradict previous data published in [7,8]? If so, they should describe it more clearly.

Conclusion

After the sentence “CSF α-synuclein levels may reflect PD pathology and rise before tests become abnormal” the authors should add the citation on :

Emamzadeh FN et al., Parkinson's Disease: Biomarkers, Treatment, and Risk Factors. Front Neurosci. 2018 Aug 30;12:612. doi: 10.3389/fnins.2018.00612.

References

Discrepancies in the bibliography should be corrected:

8. 8. Varrone A, Halldin C. Molecular imaging of the dopamine transporter. J Nucl Med. 2010 Sep;51(9):1331-4. doi: 28 10.2967/jnumed.109.065656. Epub 2010 Aug 18. PMID: 20720060.

9.

10. 9. Zhu, S.; Zhong, M.; Bai, Y.; Wu, Z.; Gu, R.; Jiang, X.; Shen, B.; Zhu, J.; Pan, Y.; Dong, J.; et al. The Association Between 31 Clinical Characteristics and Motor Symptom Laterality in Patients With Parkinson’s Disease. Front Neurol 2021, 12, 663232, 32 doi:10.3389/fneur.2021.663232.

Author Response

São Paulo, January 05th, 2023

Manuscript biomedicines-2121994

To reviewers of Biomedicines,

We would like to thank the reviewers for the careful analysis of the manuscript. We appreciate the comments, which have considerably improved our manuscript. We have addressed all the concerns raised by the reviewers with much attention. Please find below the responses to the reviewers’ comments within this letter and in the revised manuscript. The changes were added with “track changes” tool as per requested.

Sincerely,

Raquel C R Martinez, PhD

biomedicines-2121994

Title: Does TRODAT-1 SPECT uptake correlate with cerebrospinal fluid α -synuclein levels in mid-stage Parkinson’s disease?

Author’s name: Coutinho et al.

The manuscript was reviewed according to the comments and the changes are listed below:

REVIEWER 2

Parkinson’s disease is a prevalent and severe neurodegenerative disease for which currently there is no treatment modifying the course of the disorder. Furthermore, there is no reliable biomarker for early identification of the beginning of the disease, and the number of patients with the disorder is growing. Thus, this important to investigate molecular and cellular mechanisms of the disease. The authors used dopamine transporter single emission computed tomography test (DAT-SPECT) to study the imbalance in dopamine levels, and to find possible relationship with other measures of neurodegeneration, and with α-synuclein levels in the cerebrospinal fluid. This is an important field of biomedical science and the results of the manuscript will be interesting to the readers of “Biomedicines”. The following corrections and additions should be made:

  • Abstract

The following sentence should be corrected “(4) Conclusions: Our data highlight the importance of understanding the dysfunction of the dopaminergic system in the basal ganglia and its complex interactions to modulate cognition” should be corrected as follows:” (4) Conclusion: Our data highlight the importance of understanding the dysfunction of the dopaminergic system in the basal ganglia and its complex interactions in modulation of cognition”

Answer: We greatly appreciate the review for bringing this to our attention.

  • Introduction

The following sentence:” The worldwide prevalence of Parkinson’s disease (PD) has increased significantly over the last decade owing to an increase in longevity [1]” should be corrected as follows:” The worldwide prevalence of Parkinson’s disease (PD) has increased significantly over the last decade due to an increase in human longevity [1]”.

Answer: We greatly appreciate the review for bringing this to our attention.

  • After the sentence “:However, to date, its use to determine diagnosis or prognosis is still under investigation.” The authors should add a citation to a recent article:”Biomarkers in Parkinson’s Disease”. Chapter in a book Peplow P.V., Martinez B., Gennarelli T.A. (eds) Neurodegenerative Diseases Biomarkers. 2022. Neuromethods, vol 173. pp 155-180. Humana, New York, NY. https://link.springer.com/protocol/10.1007/978-1-0716-1712-0_7”

Answer: We greatly appreciate the reviewer’s advice and added this pivotal reference in our introduction.

  • “Therefore, a possible correlation between disease laterality, DAT-SPECT uptake, cognition, and CSF α-synuclein in PD is lacking” should be corrected as follows:” Therefore, a possible correlation between disease laterality, DAT-SPECT uptake, cognition, and CSF α-synuclein in PD should be thoroughly studied”

Answer: We greatly appreciate the review for bringing this to our attention.

  • “…and was inserted at the Clinical Trials: NCT02647372” should be rewritten as follows :” and was inserted at the Clinical Trials with the following ClinicalTrials.gov Identifier :NCT02647372”

Answer: We greatly appreciate the review for bringing this to our attention.

6) Results

3.1 General description of individuals with PD

“We included 28 individuals, see Figure 1 (26% female)…” The authors should explain the contradiction between 26% female shown in the text and 35.7% in the Table 1.

Answer: We are very sorry for the typing mistake. The correct percentage is 35.7, which were rounded to 36% in the text.

7) Discussion

“The present study did not confirm a dopamine imbalance between the brain hemispheres in mid-to late-stage PD [7,8].” The sense of this statement is not clear. Do the authors mean that dopamine imbalance between the brain hemispheres in mid-to late-stage PD was previously detected in these two studies? [7,8]. And their findings in this study contradict previous data published in [7,8]? If so, they should describe it more clearly.

Answer: We apologize for it. Yes, we did not find dopamine imbalance between brain hemispheres as previously shown by the authors, probably, due to longer disease duration in our cohort, as further suggested in the paragraph. We have re-written the sentence.  

8) Conclusion

After the sentence “CSF α-synuclein levels may reflect PD pathology and rise before tests become abnormal” the authors should add the citation on :

Emamzadeh FN et al., Parkinson's Disease: Biomarkers, Treatment, and Risk Factors. Front Neurosci. 2018 Aug 30;12:612. doi: 10.3389/fnins.2018.00612.

Answer: We greatly appreciate the reviewer’s suggestion and added this reference in our discussion.

9) References

Discrepancies in the bibliography should be corrected:

  1. 8. Varrone A, Halldin C. Molecular imaging of the dopamine transporter. J Nucl Med. 2010 Sep;51(9):1331-4. doi: 28 10.2967/jnumed.109.065656. Epub 2010 Aug 18. PMID: 20720060.

9.

  1. 9. Zhu, S.; Zhong, M.; Bai, Y.; Wu, Z.; Gu, R.; Jiang, X.; Shen, B.; Zhu, J.; Pan, Y.; Dong, J.; et al. The Association Between 31 Clinical Characteristics and Motor Symptom Laterality in Patients With Parkinson’s Disease. Front Neurol 2021, 12, 663232, 32 doi:10.3389/fneur.2021.663232.

Answer: We greatly appreciate the reviewer for bringing this to our attention. We have included the new references, re-checked, and made all the corrections.

Round 2

Reviewer 1 Report

The manuscript improved from the previous version and addressed all the comments.